# The Influence of Air Pollution on the Development of Allergic Inflammation in the Airways in Krakow’s Atopic and Non-Atopic Residents

**DOI:** 10.3390/jcm10112383

**Published:** 2021-05-28

**Authors:** Ewa Czarnobilska, Małgorzata Bulanda, Daniel Bulanda, Marcel Mazur

**Affiliations:** 1Department of Clinical and Environmental Allergology, Jagiellonian University Medical College, Botaniczna St. 3, 31-501 Krakow, Poland; ewa.czarnobilska@uj.edu.pl (E.C.); malgorzata.l.lesniak@gmail.com (M.B.); 2Department of Biocybernetics and Biomedical Engineering, AGH University of Science and Technology, Mickiewicza Av. 30, 30-059 Krakow, Poland; kbib@agh.edu.pl

**Keywords:** air pollution, respiratory health, allergic inflammation

## Abstract

Until now, the simultaneous influence of air pollution assessed by measuring the objective marker of exposition (1-hydroxypirene, 1-OHP) and atopy on the development of allergic airway diseases has not been studied. The aim of this study was to determine the pathomechanism of the allergic response to PM2.5 in atopic and non-atopic patients. We investigated the changes in peripheral blood basophil activity of patients after stimulation with the birch pollen allergen alone, the allergen combined with PM2.5 (BP), PM2.5 alone, a concentration of 1-OHP in urine, and a distance of residence from the main road in 30 persons. Activation by dust alone was positive for all concentrations in 83% of atopic and 75% of non-atopic assays. In the group of people with atopy, the simultaneous activation of BP gave a higher percentage of active basophils compared to the sum of activation with dust and birch pollen alone (B + P) for all concentrations. The difference between BP and B + P was 117.5 (*p* = 0.02) at a PM concentration of 100 μg. Such a relationship was not observed in the control group. The correlation coefficient between the distance of residence from major roads and urinary 1-OHP was 0.62. A Pearson correlation analysis of quantitative variables was performed, and positive correlation results were obtained in the atopy group between BP and 1-OH-P. Exposure to birch pollen and PM2.5 has a synergistic effect in sensitized individuals. The higher the exposure to pollutants, the higher the synergistic basophil response to the allergen and PM in atopic patients.

## 1. Introduction

Presently, allergies are a public health problem with a pandemic range, especially in industrialized countries. Globally, 400 million people suffer from rhinitis. According to the World Health Organization, the number of patients suffering is 300 million, and with the rising trends, this number is expected to increase to 400 million by 2025. One of the possible factors explaining this phenomenon is growing up and living in conditions of increased air pollution (AP) [1,2]. AP relates to substances emitted above permissible levels in ambient air. According to the available reports, the air quality in the Kraków Agglomeration exceeds the established European concentration norms defined for particulate matter PM10 and PM2.5, as well as benzo[a]pyrene in PM. The main reason for exceedances is surface emission connected with the individual heating of buildings in the communal and household sector, as well as transport emission. A significant share of air pollution in Kraków comes from the inflow from the neighboring communes [3].

Coarse PM, with an aerodynamic diameter of 2.5–10 μm, is mainly deposited in the nose and large conducting airways. Fine PM or PM2.5 deposits particularly in small airways and alveoli. Various individual factors modulate airway distribution and the depositing of PM, such as maturation and integrity of the respiratory tract, general health, and nasal versus oral breathing. Polycyclic aromatic hydrocarbons (PAH), transition metals, and bacterial lipopolysaccharides are constituents of PM of special interest because of their potential to cause oxidative stress, apoptosis response, airway hyper-responsiveness, and airway remodeling [4,5]. Benzo[a]pyrene is a representative of the PAH group. There are some methods of measurement of exposition to PAHs using biomarkers. The most commonly used is 1-hydroxypyrene (1–0HP), the main metabolite of pyrene, which correlates with a pyrene concentration in the air and breathing fraction of PM [6].

An in-depth analysis of the impact of PM exposure on health is still inconclusive based on data from experimental or epidemiological studies. Some studies have investigated the effect of natural exposure to PM2.5 on the health of traffic policeman or employees at trucking terminals, but without an explanation of the mechanism [7,8]. There is a growing body of literature on the mechanistic effect of pollutants on atopic and non-atopic individuals. In general, environmental and occupational pollutants irritate the nasal and bronchial mucosa, leading to the release of inflammatory mediators, which overlaps with the symptomatology of atopic diseases, like allergic rhinitis (AR) or asthma. Pollution can involve the ocular system, although inconsistently and non-specifically. Not all studies are supportive of such pollutant-induced, non-allergic inflammation [5].

Atopic individuals can have different responses to pollutants compared to a non-atopic population. The role of air pollutants in triggering allergies perhaps depends both on the adjuvant effect at the stage of causing sensitization and allergenic provocation leading to the development of an allergic disease. It is now believed that allergens in combination with AP have a greater sensitizing potential [9].

For ethical reasons, it is impossible to perform in vivo provocations with PM in humans, and most studies on its mechanism of action are performed in mice [10]. The basophil activation test (BAT) is an in vitro specific provocation test, which can replace potentially dangerous in vivo provocations [11].

Until now, the simultaneous influence of air pollution assessed by measuring the objective marker of exposition (1-hydroxypirene (1-OHP)) and atopy on the development of allergic airway diseases in the inhabitants of Krakow has not been studied. The aim of this study was to determine the pathomechanism of the allergic response to PM2.5 in adult residents of Kraków. Under experimental conditions, BAT was performed in atopic and non-atopic patients after stimulation with PM2.5 in the presence or absence of birch allergen to see which group is more sensitive.

## 2. Materials and Methods

### 2.1. Patients

The study involved 30 patients referred to the Allergy Clinic of the Clinical and Environmental Allergy Center of the University Hospital in Krakow due to suspicion of AR in the winter season, meeting restrictive study inclusion criteria. An allergy interview was collected during the qualification visit, including the question of the distance of residence from the main road (below or above 300 m) and smoking. Skin prick tests (SPTs) with a set of basic inhalant allergens (D. pteronyssinus, D. farinae, cat, dog, grass, cereals, hazel, alder, birch, mugwort, Alternaria tenuis) were performed. Additionally, exfoliative cytology performed in all of the 30 patients confirmed eosinophilic inflammatory infiltrate of the nasal mucosa (percentage of eosinophils among influx cells was greater than 2%). Based on the results of the above tests, patients were assigned to two groups:symptomatic (with atopy and an allergy to birch, no allergy to all-year allergens, non-desensitized);control group (with symptoms, without atopy).

The exclusion criteria for the study group allowed for reliable BAT results to be achieved during winter (December/January). The study was approved by the regional scientific ethics committee.

### 2.2. PM Preparation

The dust was obtained from the Voivodship Inspectorate for Environmental Protection measuring station in Kraków-Kurdwanów, where it was deposited on a porous cellulose filter (daily filters from the 30-day period in December and January). Dust was separated from the carrier by mechanical separation with the use of an ultrasound, in combination with an inert liquid (isopropanol). This standard liquid was verified in each case, taking both the physical dissolution of certain organic and inorganic substances, as well as the possibility of chemical reactions, into account.

### 2.3. Basophil Activation Test

The assessment of basophil activation by measuring CD63 antigen expression was carried out by means of the Flow2 CAST test (Bühlmann Laboratories AG, Schönenbuch, Switzerland). Anti-FcɛRI monoclonal antibodies (antibodies against the high-affinity IgE receptor) were used, and a positive control and stimulation buffer were used as a negative control. Peripheral blood basophils were isolated from whole peripheral blood. Granulocytes were enriched via density gradient centrifugation and processed to remove remaining red blood cells. Isolated cells were characterized by flow cytometry to ensure a highly pure and viable cell population. Basophils were stimulated with original allergen solutions (Bühlmann Laboratories AG, Schönenbuch, Switzerland) in one concentration recommended by the manufacturer: 22.5 ng/mL and/or solutions of PM2.5 in five concentrations: 100 μg/mL, 75 μg/mL, 50 μg/mL, 25 μg/mL, and 12.5 μg/mL (selected in the preliminary experience). A mixture of FITC-conjugated anti-CD63 monoclonal antibodies and PE-conjugated anti-CCR3 antibodies were added to the cellular suspension, and were incubated for 15 min at 37 °C, after which 2 mL of erythrocyte lysing solution was added. The samples were centrifuged, resuspended in 0.3 mL of washing buffer, and analyzed within one hour on a flow cytometer FACS Canto II (BD, Biosciences) with basophils gated as CCR3 +/SSC low cells. The test results were presented as the percentage of activated basophils and stimulation index (SI). SI determines the percentage of basophils activated with an appropriate allergen in relation to the number of basophils activated in the negative control, and is used to describe the reactivity of patients to the allergen. SI ≥ 2 is deemed as a positive result [11].

### 2.4. 1-OHP Assessment

1-OHP in urine was determined by high-performance liquid chromatography with fluorescence detection. The assay process involved the following steps: enzymatic hydrolysis of urine, extraction with ethyl acetate, chromatography resolution on an HPLC column Kinetex 5 μ C18 100A (150 × 4.6 mm) in gradient conditions, and fluorescence detection λ Ex/λ Em (280/390) nm. Chemicals: acetate buffer (pH = 5.5), β-glucuronidase/arylsulphatase, acetonitrile HPLC gradient grade, deionized water, methanol HPLC gradient grade, ethyl acetate, and 1-OHP standard. Equipment (onsite): liquid chromatograph (HPLC) with a LaChrom (Merck-Hitachi High Tech, Tokyo, Japan) fluorescence detector; other minor equipment: heater, thermo-block, laboratory centrifuge, analytical balance, micropipettes for various volumes of 10 μL to 5000 μL. Urine creatinine was the parameter needed for the standardization of urine 1-OHP results. Urine creatinine was measured by an enzymatic assay (EMIT) using a reagent kit immunoanalyzer (Siemens Healthcare Diagnostics Inc., Tarrytown, NY, USA). According to Jongeneelen et al., the threshold of 1-hydroxypyrene in urine is 0.46 μg/g [12] for non-occupationally exposed, non-smoking individuals.

### 2.5. Statistical Analysis

BAT results generally obeyed the normal distribution. Therefore, the parametric methods were preferred for data analysis, namely the t-test for difference checks and linear regression analysis for variables dependencies. Correlation coefficients were calculated using the Pearson test. Statistical analyses were conducted using Julia v. 1.5.3 (open source (https://github.com/JuliaLang/julia), MIT License; accessed on 24 May 2021)) and Microsoft Excel for Windows (Microsoft Corporation, Redmond, WA, USA).

## 3. Results

The clinical characteristics of patients in the atopic group are shown in Table 1, and those in the non-atopic group in Table 2.

In the atopic group, the mean, minimum, median, and maximum values of basophil activation after stimulation with PM2.5 for a concentration of 100 μg were 8.40%, 1.16%, 6.91%, and 19.07%, and in the non-atopic group, they were 8.84%, 1.10%, 8.16%, and 24.54%, respectively. Basophil activations after stimulation with the other tested dust concentrations and combined birch allergen and dust (BP) for both groups are summarized in Table 3 and Table 4.

Activation by dust alone as measured by SI was positive for all concentrations in 83% of atopic and 75% of non-atopic assays.

The student’s t-test was used to check the differences of SI for simultaneous stimulation with birch allergen and dust (BP) and the sum of single stimulations with birch plus PM2.5 (B+P) for successive concentrations. In the atopic group, statistically significant differences were obtained between BP and B+P: at a PM concentration of 100 μg, the difference was 117.5, which was 263% higher (*p* = 0.021903894); at a concentration of 75 μg, the difference was 117.08, which was 274% higher (*p* = 0.026599013); at a concentration of 50 μg, the difference was 118.41, which was 290% higher (0.025584798); at a concentration of 25 μg, the difference was 112.41, which was 286% higher (*p* = 0.028641198); and at a concentration of 12.5 μg, the difference was 108.74, which was 283% higher (*p* = 0.030959045). No statistically significant differences were obtained in the non-atopic group. Graphs for both groups and each concentration are shown in Figure 1A–E.

The results of measuring 1-OHP concentrations for the atopic group are shown in Table 1 and for the non-atopic group in Table 2; 1-OHP concentrations exceeded 0.46 μg/g in 10 patients in the atopic group and five patients in the non-atopic group.

The correlation between the distance of residence from major roads and urinary 1-OHP concentrations was examined, yielding a correlation coefficient of 0.62. The mean 1-OHP concentration in the group living < 300 m from a road was 1.06 μg/g, whereas, in the group living > 300 m, it was 0.20 μg/g, as shown in Figure 2.

A Pearson correlation analysis of quantitative variables was performed, and positive correlation results were obtained in the atopy group between BP and 1-OHP, while in the other variants, i.e., the atopy group stimulated with PM alone, as well as the non-atopy group stimulated with PM and BP, the correlation was negative. Statistically significant results in the atopic group when stimulated with PM alone were obtained for a concentration of 100 μg (*p* = 0.041602, linear correlation coefficient minus 0.53), as well as in the non-atopic group for both PM alone at a concentration of 100 μg (*p* = 0.025095, linear correlation coefficient minus 0.57) and BP at a concentration of 100 μg (*p* = 0.04729058, linear correlation coefficient minus 0.54). Graphs of these correlations are shown in Figure 3A–D.

## 4. Discussion

Exposure to PM2.5 is closely related to acute and chronic diseases [10]. In our study, PM2.5 alone was positive for basophil activation in 83% of atopic and 75% of non-atopic patients in Kraków. It is worth noting that all patients, both atopic and non-atopic, had symptoms suggestive of allergic respiratory diseases during the winter season, and none of them were allergic to year-round allergens, including house dust mites. For 10 years, a preventive check study was carried out in Kraków as part of the “Healthy Kraków” City Health Care Program; 75,000 respondents were surveyed in the study. Over 50% of the respondents reported allergic symptoms, and the atopic background was confirmed only in 50% of the participants reporting symptoms of respiratory allergies [13]. It may be that the exposure to AP, including PM2.5, is the cause of symptoms in a significant proportion of these individuals [14]. For ethical reasons, most experimental studies on the effects of PM2.5 are not conducted on patients. The work of Murugappan et al. showed that airborne PM induces non-allergic eosinophilic sinonasal inflammation in mice. The discussed study demonstrates the destructive effects of chronic airborne PM on sinonasal health in vivo, including the epithelial barrier breakdown, proinflammatory cytokine release, inflammatory cell accumulation, and eosinophilic inflammation [15]. PM intensifies Th2 and Th17 phenotypic differentiation, with a specific role for environmentally persistent free radicals and polycyclic aromatic hydrocarbon fractions. Several studies have reported changes in gene expression following PM2.5 exposure [16,17]. Work by Ouyang Y. et al. on a mouse model suggest a mechanism by which PM2.5 in ambient air pollution may stimulate the innate immune system through the PM2.5-Nod1-NF-κB axis in chronic allergic disease [10]. In a pilot study conducted on 11 patients, stimulation with organic extracts of urban aerosol alone (AER) exhibited no effects on CD63 expression in basophils from atopic donors, as well as from healthy controls. It is worth noting that the patients analyzed by these investigators did not have symptoms of allergic airway disease during the winter season. This may explain the negative BAT with AER in the entire 11-person group and the positive BAT with PM2.5 in most of the symptomatic patients we studied [18].

Another important finding of our study is the synergistic effect of simultaneous stimulation with birch allergen and PM2.5 in atopic patients. This relationship was not observed in non-atopic patients. This confirms that individuals can have different responses to pollutants compared to a healthy population [5]. In vivo repeated exposure to allergens, which would occur during seasonal exposure, leads to increasing tissue inflammation, which may last for days. The inflammation changes the reactivity of the nasal mucosa to further exposure to allergens and nonspecific irritants [19]. In our in vitro model, basophils sensitized simultaneously with a sensitizing allergen and a nonspecific antigen responded with statistically significantly higher activity than would be indicated by the arithmetic sum of individual B + P stimulations. This may suggest an important role of PM in the aggravation of IgE-mediated allergic diseases [17].

Several mechanisms through which air pollutants could enhance sensitization to aeroallergens have been proposed, and include the increased deposition of allergens in the airways due to carriage by particles, increased epithelial permeability due to oxidative injury, and increased antigenicity of proteins from chemical modification [20]. PM can act as adjuvants and promote long-lasting allergic inflammation in the airways. The mechanism may involve the killing of alveolar macrophages, leading to the release of IL-1α and the formation of inducible bronchus-associated lymphoid tissue in the lungs [21]. This is of specific significance, as exposure to natural inhalant allergens, mainly tree pollen, often coincides with a high concentration of PM, as confirmed by analyses carried out in Kraków. A high concentration of birch pollen and an exceeded admissible level of PM concentration were found in more than 60% of the days in the city center. In 70.3% of the days in which the birch pollen concentration reached values that caused the symptoms of asthma (>155 pg/m^3^), the dust concentration was exceeded (> 50 µg/m^3^), and rainfall and wind speed of 1–3 m/s occurred [22].

In our patients, we additionally performed urinary 1-OHP determinations as a marker of exposure. The biological half-life of 1-OHP ranges from 6 to 35 h, with an average of 18 h [23]. The presence of 1-OHP in urine in subjects not exposed to tobacco smoke is a confirmation of environmental exposure to beznzo(a)pyrene. This is confirmed by the relationship we obtained between the distance from the road and the urinary 1-OHP concentration. The 1-OHP concentrations exceeded 0.46 μg/g in 10 patients in the atopic group and five patients in the non-atopic group.

In the analyzed group of patients, we have shown the correlation between the distance of residence from main roads and the concentration of 1-OHP in urine. Studies conducted earlier in Kraków, Poland, have shown that children and adolescents with a residential proximity closer to a major roadway had more frequent asthma-related symptoms. Respondents residing within 200 m complained more often of sneezing and a runny or blocked nose accompanied by itchy, watery eyes and hay fever in comparison to respondents who resided 200–500 m from a major roadway [24]. Concentrations of many of the constituent pollutants in AP diminish quickly with the distance from the roadways. A 2010 review suggested that distances within 300–500 m of roadways were the most relevant for effects on human health [25].

We also obtained a positive correlation between basophil activity after stimulation with birch allergen and PM2.5 at a concentration of 100 μg and 1-OHP concentration in atopic patients. The higher the exposure to air pollutants measured with 1-OHP, the higher the synergistic basophil response to allergen (birch pollen) and PM in atopic patients. However, we assume that birch pollen exposure is mainly responsible for the observed effect. Clinically, this may account for the more severe allergy symptoms associated with exposure to sensitizing birch pollen in polluted air. Meanwhile, in the non-atopic and atopic group stimulated with particulate matter alone, we observed a negative correlation between basophil activation and 1-OHP concentration. We can speculate that exposure to benzo[a]pyrene reduces the reactivity of cells involved in the immune response, including basophils. This certainly requires further study.

Our results have important clinical implications. Management of atopic patients should include not only treating symptoms according to standards for AR and/or asthma, but also avoiding exposure to air pollutants. They should follow messages about pollution levels and avoid unnecessary outdoor activities when standards are exceeded. These recommendations should be added to the treatment plan of patients with allergic diseases [4]. It is possible that ineffective treatment, including the incomplete effect of specific immunotherapy in some patients with tree pollen allergies, is related to concurrent exposure to allergens and pollutants. Furthermore, non-atopic patients with respiratory symptoms after exposure to air pollutants should avoid exposure. With symptoms of allergic airway disease during the heating season and negative tests for natural environmental allergens, the sensitizing effect of AP should be considered.

In conclusion, the strengths of our study are the performance of specific in vitro provocations with PM2.5 at five different concentrations and the measurement in urine of the biological exposure marker 1-OHP. By performing the in vitro challenge with PM2.5 only, we eliminate the influence of other AP, both intrinsic and extrinsic, on the challenge outcome and demonstrate the mechanism of the allergic effect of PM2.5 through the activation of basophils and increased expression of CD63 on their surface. The measurement of 1-OHP illustrates the exposure to benzo[a]pyrene, which is measured as a component of PM in Kraków. A weakness is the small size of the study group, but it is representative and sufficient to obtain statistically significant results, and we consider it as a pilot group. Since none of our patients were sensitized to natural year-round allergens, and all of them had confirmed eosinophilic nasal mucositis and demonstrated symptoms of allergic respiratory tract diseases, it is highly likely that exposure to AP, including PM2.5, was the provoking factor for these symptoms. This issue requires further study.

## 5. Conclusions

1.Particulate matter (PM) 2.5 in atopic and non-atopic people activates basophils of peripheral blood and causes the expression of CD63 receptors on their surface. Dust alone activates basophils in 83% of determinations in atopic people and 75% of determinations in non-atopic people in Kraków.2.Exposure to birch pollen and PM2.5 has a synergistic effect in sensitized individuals.3.The higher the exposure to pollutants, the higher the synergistic basophil response to the allergen and PM in atopic patients.

## Figures and Tables

**Figure 1 jcm-10-02383-f001:**
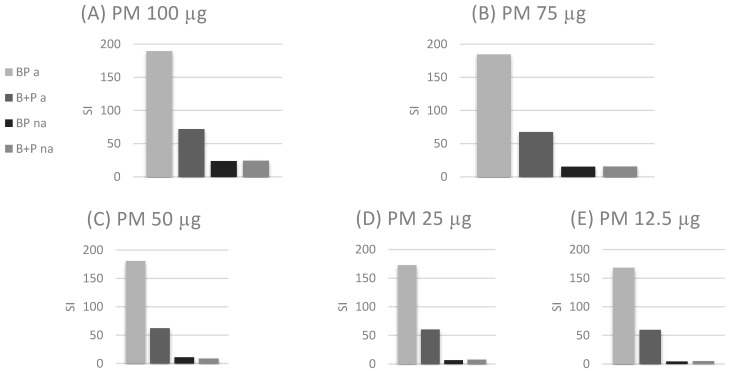
Differences of stimulation index (SI) for simultaneous stimulation with birch allergen and dust (BP) and the sum of single stimulations with birch pollen and PM2.5 (B + P) for successive concentrations in atopic and non-atopic individuals. (**A**) PM 100 µg; (**B**) 75 µg; (**C**) 50 µg; (**D**) 25 µg; (**E**) 12.5 µg. The meaning of abbreviations: BP a—simultaneous stimulation with birch allergen and dust in the atopic group; B+P a—the sum of single stimulations with birch allergen and dust in the atopic group; BP na—simultaneous stimulation with birch allergen and dust in the non-atopic group; B+P na—the sum of single stimulations with birch allergen and dust in the non-atopic group.

**Figure 2 jcm-10-02383-f002:**
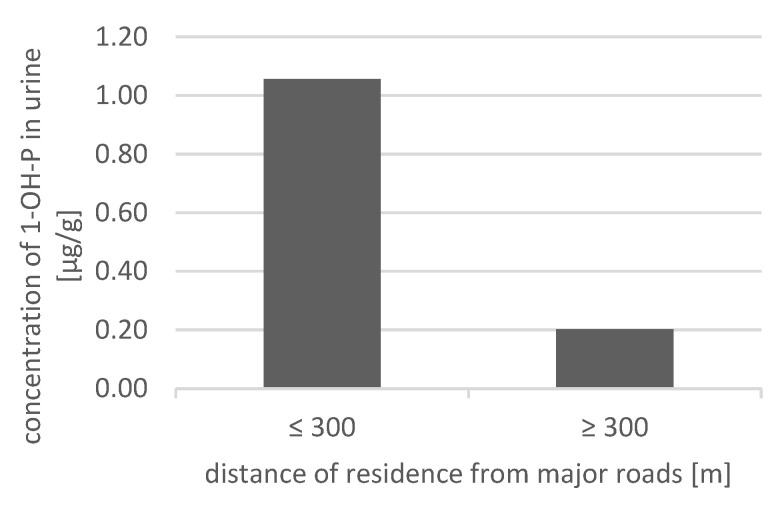
The concentration of 1-OHP in urine in two different groups according to distance of residence from major roads.

**Figure 3 jcm-10-02383-f003:**
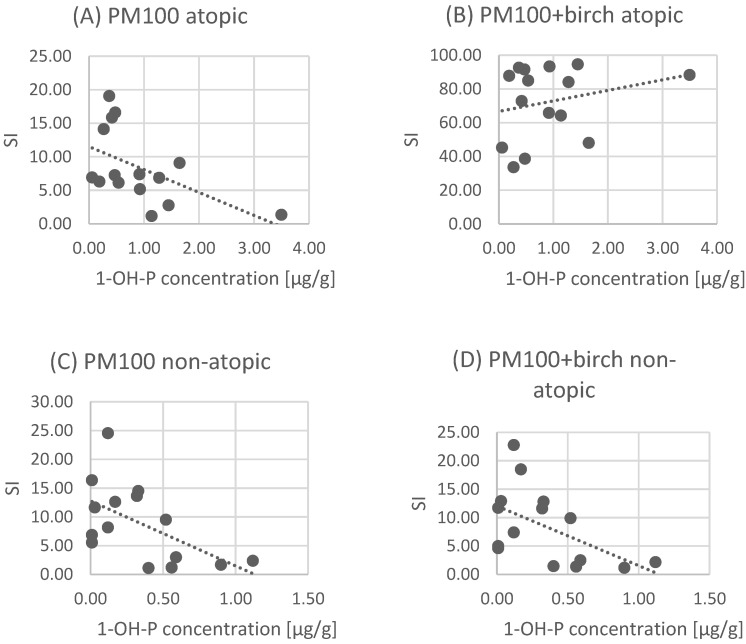
Correlation analysis of stimulation index (SI) and 1-OHP concentration in examined groups. (**A**) PM 100 atopic; (**B**) PM 100 + birch atopic; (**C**) PM 100 non-atopic; (**D**) PM 100 + birch non-atopic; Each data point represents 1-OH=P and SI correlation for one patient. The dotted lines represent the fitted linear function for a given set of data points.

**Table 1 jcm-10-02383-t001:** The clinical characteristics of patients in the atopic group.

	Age	Sex	Symptoms	BAT with Birch [%]	1-OHP in Urine [μg/g]
1	19	F	AR	83.17	0.47
2	48	F	AR + Asthma	40.2	1.14
3	68	F	AR	80.21	3.50
4	49	M	AR	55.79	1.28
5	47	F	AR	42.62	0.92
6	65	F	AR	83.00	1.45
7	59	M	AR + Asthma	33.78	1.65
8	56	F	AR + Asthma	54.73	0.42
9	27	M	AR	29.09	0.06
10	47	F	AR	76.73	0.37
11	20	F	AR + Asthma	14.82	0.27
12	67	M	AR	81.30	0.19
13	70	F	AR	16.00	0.48
14	40	F	AR	84.40	0.93
15	47	M	AR + Asthma	76.08	0.54

**Table 2 jcm-10-02383-t002:** The clinical characteristics of patients in the non-atopic group.

	Age	Sex	Symptoms	BAT with Birch	1-OHP in Urine [μg/g]
1	51	M	AR + Asthma	1.86	0.12
2	46	F	AR + Asthma	0.65	1.12
3	41	F	AR	2.18	0.01
4	18	F	AR	2.66	0.52
5	18	M	AR	1.31	0.33
6	60	F	AR + Asthma	1.69	0.32
7	56	F	AR	0.20	0.90
8	35	M	AR + Asthma	0.40	0.59
9	42	F	AR	0.90	0.40
10	38	F	AR	1.22	0.17
11	39	F	AR	0.50	0.03
12	55	F	AR	0.53	0.01
13	44	M	AR	1.49	0.01
14	29	F	AR	0.42	0.12
15	62	F	AR	0.57	0.56

**Table 3 jcm-10-02383-t003:** Basophil activations after stimulation with PM2.5 and combined birch allergen and dust (BP) in the atopic group.

	Average [%]	Minimum [%]	Median [%]	Maximum [%]
PM 100 μg	8.40	1.16	6.91	19.07
PM 75 μg	6.96	0.64	5.25	17.81
PM 50 μg	3.85	0.59	2.91	10.39
PM 25 μg	2.54	0.36	1.93	7.19
PM 12.5 μg	1.94	0.37	1.20	4.74
BP 100 μg	72.41	33.65	84.09	94.60
BP 75 μg	69.66	29.80	77.59	93.50
BP 50 μg	68.12	27.30	70.66	94.20
BP 25 μg	64.45	22.10	68.41	92.80
BP 12.5 μg	61.79	21.00	60.64	88.50
negative control	0.91	0.10	0.70	2.77

**Table 4 jcm-10-02383-t004:** Basophil activations after stimulation with PM2.5 and combined birch allergen and dust (BP) in the non-atopic group.

	Average [%]	Minimum [%]	Median [%]	Maximum [%]
PM 100 μg	8.84	1.10	8.16	24.54
PM 75 μg	6.27	0.77	5.52	17.60
PM 50 μg	4.38	0.54	2.60	11.80
PM 25 μg	3.47	0.24	1.85	9.70
PM 12.5 μg	2.03	0.32	1.47	6.03
BP 100 μg	8.38	1.17	7.37	22.75
BP 75 μg	6.96	0.77	5.08	19.10
BP 50 μg	4.62	0.62	4.07	10.80
BP 25 μg	3.12	0.56	2.14	8.35
BP 12.5 μg	2.13	0.46	1.83	5.14
negative control	0.87	0.08	0.54	2.51

## Data Availability

The data presented in this study are available on request from the corresponding author. The data are not publicly available due to General Data Protection Regulation.

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
