# Peer review of "The Influence of Air Pollution on the Development of Allergic Inflammation in the Airways in Krakow’s Atopic and Non-Atopic Residents"

_jcm, 2021, doi:10.3390/jcm10112383_

Round 1

Reviewer 1 Report

To the authors.

The manuscript presents a very interesting study on the potential contribution of air pollution to allergic disease. The study is well designed and the experiments well executed and analysed. The conclusions, while acknowledging the limitations of the study, have potentially useful implications in the clinical management of patients with allergic disease, such as taking into account distance of residence of the patient from major roads.

I have some comments and recommendations in the methods and results sections:

It would benefit the reader if the methods section was organised into subsections, each describing a technique or experimental procedure, eg. Patient information collection, PM collection/preparation, Basophil activation assessment, Urine analysis etc.

Line 105: ‘cells were stimulated..’ Please provide more information on the in vitro experiments and clarify what cells were used in these experiments, how they were isolated and treated.

There is some confusion in the description of the patient groups (line 88-91), the experimental groups should be more clearly defined. Also, there are similar discrepancies in the acronyms used for birch allergen and PM stimulation between the results text and some figures (B, BP, B+P, Tables 3 and 4 and lines 155-157. Please use a uniform and consistent way of describing the different conditions.

Thank you.

Author Response

Thank you for your valuable comments increasing the value of the submitted paper. The authors have introduced the suggested corrections to the text. The corrected paper has been reuploaded.

Reviewer 2 Report

In this research article, the authors present an in vitro study based on basophil activation tests where they show a synergistic effect of PM2.5 and allergens from birch pollen in atopic patients in a small cohort of residents of the area of Krakow. The authors also show a positive correlation between the exposure to pollution and the stimulation index of basophils treated with the combination PM2.5 and birch pollen in the atopic group.

The global message of the manuscript is clear and brings solid pieces of evidence and, the discussion is clear and well-argued. Additionally, the authors are aware of the limitations of the study. However, the following minor comments need to be addresed for the sake of a better understanding:

Mayor comments:

  1. The authors agree that the main weakness of the study is the size of the study population however, they state that the sample is representative and sufficient to obtain statistically significant results. Based on what is this affirmation made?
  2. Institutional Review Board Statement section is needed.

Minor comments:

Line 20: sentence in the abstract must be corrected.

Line 36: sentence must be rephrased to clarify how the air quality of Krakow is.

Line 60: abbreviations must be explained the first time used (AR=allergic rhinitis)

Figure 1: the legend must be clear with what BA a, B+P a, BP na and B+P na. 

Line164: figure 1 does not reflect A to E legends.

Figure 2: It represents the concentration of 1-OH-P in two different groups, not correlation. Figure legend must be corrected.

Figure 3: No units are shown in the X and Y axes of the different graphs. It must be corrected. All graphs must include 

Line 189: Graphs of figure 3 are not named A to D.

Funding and Institutional Review Board Statement sections are not filled in.

Author Response

(The authors gave the same response as above.)
